## Research Article

mental health; whole family intervention; adolescent girls; migrants and returnees

**Corresponding author:**
Lindsay Stark;
Email: lindsaystark@wustl.edu

# Feasibility, acceptability and implementation of a whole-family mental health intervention for displaced adolescent girls in Colombia: A mixed-methods pilot randomized controlled trial

Ilana Seff[1] ⬥, Arturo Harker Roa[2,3], Byron Powell[1], Natalia Cordoba[4],
Carolina Rodriguez[5], Julianne Deitch[6], Elvia Tamaity Ariza Pena[5],
Feven Gebrekidan[1] and Lindsay Stark[1] ⬥

[1]Washington University in St Louis, USA; [2]School of Government, Universidad de los Andes, Colombia; [3]Imagina Research Center, Universidad de los Andes, Colombia; [4]Universidad de los Andes, Colombia; [5]Mercy Corps, Colombia and [6]Women's Refugee Commission Inc, USA

## Abstract

Adolescent girls affected by displacement face substantial mental-health risks. The Sibling Support for Adolescent Girls in Emergencies (SSAGE) is a 12-week, gender-transformative, family-based program designed to improve adolescent girls' mental health in humanitarian settings. This mixed-methods pilot randomized controlled trial (RCT) assessed SSAGE's feasibility, acceptability and potential effects among 186 Venezuelan migrant and Colombian returnee families in Colombia. Adolescent girls aged 13–19 years, their male siblings and caregivers participated in parallel sessions on gender dynamics, communication and relationships. Implementation outcomes drew on the Mental Health Implementation Science Tools (acceptability and feasibility subscales), attendance records and qualitative interviews. Analyses followed an intent-to-treat approach using adjusted linear and logistic regression models. Quantitative analyses did not identify measurable changes in adolescent girls' mental health outcomes at endline; however, attendance was modest, with only ~10% of families meeting the predefined protocol threshold. Implementation findings revealed strong participant satisfaction and high acceptability of SSAGE content and mentor relationships. Engagement was constrained by economic hardship, transportation and venue barriers, and some caregivers' acute emotional distress, which likely limited feasibility and potential impact. SSAGE shows promise as a gender-transformative, family-based approach, but successful delivery in urban migrant settings will require tailored and refined implementation strategies.

## Impact statement

Few interventions work with whole families to shift the gender norms that contribute to increased mental health risks for forcibly displaced adolescent girls. This pilot randomized controlled trial study assessed the Sibling Support for Adolescent Girls in Emergencies program's feasibility, acceptability and potential effects for Venezuelan migrant and Colombian returnee families in Colombia. Data show strong programming acceptability and satisfaction, but low attendance and high barriers to engagement, potentially limiting feasibility and potential impact. Findings contribute to the emerging evidence base on gender-transformative whole-family approaches in humanitarian settings and offer practical insights for future programming and research to support displaced adolescent girls.

## Background

The number of forcibly displaced people worldwide has significantly increased over the past decade to ~123.2 million (UNHCR, 2025a). This population faces significant mental health challenges due to exposure to violence, loss of social networks and the psychological strain of adapting to new environments (World Health Organization, 2025). The disparate nature of these mental health challenges necessitates targeted interventions to address this growing crisis (World Health Organization, 2025).

In Colombia, the mental health burden is exacerbated by the country's history of continued armed conflict and the recent influx of Venezuelan migrants. Colombia hosts nearly 7 million internally displaced persons (IDPs) and over 2.5 million Venezuelan migrants fleeing economic and political turmoil (UNHCR, 2025b). Both groups experience elevated rates of mental health

disorders: studies find that 16.4% of IDPs in Colombia self-report symptoms suggestive of psychological distress, while 58.7% of Venezuelan migrants in the Colombian border area report multiple emotional/psychological symptoms, such as sleeplessness (49.0%), fright (54.4%) and tiredness (39.8%) (Tamayo Martínez et al., 2016; Bautista et al., 2025). These challenges are compounded by socioeconomic stressors, including poverty, discrimination and lack of access to mental health services (Alarcon et al., 2022; World Health Organization, 2025). Among these groups, adolescent girls are particularly vulnerable, facing unique risks due to their age and gender.

The mental health challenges of forcibly displaced adolescent girls in Colombia are both urgent and under-addressed. Although this population may exhibit profound resilience, displacement in this setting deepens structural gender-based violence and inequality, compounding existing emotional health challenges for women and girls specifically (Zamora-Moncayo et al., 2021). A 2015 National Mental Health Survey found comparatively higher rates of mental health disorders among displaced adolescents, including anxiety, depression, post-traumatic stress disorder and suicidal thoughts and attempts, *versus* their nondisplaced peers (Marroquín Rivera et al., 2020). Globally, scholars have similarly observed that displaced adolescent girls resettled in high-income countries face higher rates of psychological distress due to trauma, acculturative stress and family separation (Fazel et al., 2012). In low- and middle-income host countries, barriers to mental health care further exacerbate these challenges, with violence exposure identified as a key risk factor and stable settlement and social support as critical protective factors for mental health issues (Reed et al., 2012).

Gender norms and family structures play a pivotal role in shaping the mental health outcomes of displaced adolescent girls (Stark et al., 2018, 2025). Traditional gender roles may burden girls with disproportionate caregiving responsibilities, limiting their access to education, employment and social connection, thereby heightening stress and hindering personal development (Baird et al., 2019; Campbell et al., 2021; Seff et al., 2021). Family dynamics can either buffer or worsen the effects of displacement: supportive environments can promote resilience, while dysfunctional relationships may exacerbate psychological distress (Fazel et al., 2012; Reed et al., 2012; Tamayo-Aguledo et al., 2022). In light of these dynamics, gender-transformative whole-family interventions have emerged as a promising strategy (Meinhart et al., 2024). These programs engage all family members to challenge restrictive gender norms and foster equitable relationships, aiming to cultivate environments that support girls' mental health and well-being (Seff et al., 2024). However, the evidence base for such interventions in humanitarian settings remains limited, necessitating further research on their effectiveness (Meinhart et al., 2024).

Further, despite growing recognition of the mental health needs of forcibly displaced populations, far less attention has been paid to *how* interventions are implemented in humanitarian and low-resource settings, and why promising programs often fail to achieve impact at scale. Implementation science scholarship from low- and middle-income countries (LMICs) emphasizes that intervention effectiveness is shaped not only by intervention content, but by delivery strategies that are adaptable to the unique health system constraints, cultural contexts and lived experiences of local populations. As Alonge and Brooks (2025) argue, traditional implementation frameworks derived from high-income settings often fail to address the realities of LMICs, and adaptation for local contexts is essential to avoid perpetuating inequities and achieve real-world impact. In humanitarian settings in particular, economic precarity, mobility constraints and nontraditional caregiving responsibilities, among other factors, can substantially influence feasibility, fidelity and sustained engagement (Sánchez-Céspedes, 2017; Shenderovich et al., 2018; Peycheva et al., 2023). Interventions designed without centering participant voices risk overlooking contextual dimensions that shape how programs are received and enacted in practice (Chase and Mosse, 2025); and, further, failure to evaluate the implementation of these interventions risks losing vital insights on how to ensure greater impact in the future.

### The Sibling Support for Adolescent Girls in Emergencies (SSAGE) intervention

SSAGE is a 12-week, gender-transformative intervention that adopts a whole-family model to promote the mental health and psychosocial well-being of adolescent girls in humanitarian settings, directly addressing the interconnected risk factors facing displaced adolescent girls. As originally designed, each participating family includes an adolescent girl, an adolescent male relative in the household, a male caregiver and a female caregiver. These individuals engage in parallel, in-person, age- and gender-tailored sessions – each, ~1.5–2 h – designed to encourage critical reflection and open dialog on topics such as gender dynamics, power, effective communication and healthy relationships (Supplement A). While formal program sessions are age- and gender-disaggregated, program mentors encourage family-wide discussions outside of the program to reinforce key learnings across participants. By addressing multiple relational domains – spousal, caregiver-child and sibling – SSAGE fosters more supportive and equitable family environments. The intervention is designed to simultaneously strengthen girls' social support networks and reshape family gender norms through family member engagement. Additional details on the intervention's protocol and theory of change have been published elsewhere (Seff et al., 2024).

SSAGE was originally designed for implementation with internally displaced populations in Northeast Nigeria and was subsequently adapted for use with Syrian refugees in Jordan and internally displaced populations and refugees in Niger (Seff et al., 2022, 2023). Qualitative and mixed-methods evaluations of these implementations document improvements in family functioning, gender-equitable attitudes and sibling communication, alongside gains in girls' self-confidence, agency and psychosocial well-being (Koris et al., 2022, 2023; Seff et al., 2023). These studies also highlight important implementation lessons, including the need for careful facilitation of protection messaging to avoid reinforcing restrictive gender norms and the importance of contextual adaptation through participatory processes. Together, this body of work demonstrates SSAGE's feasibility, acceptability and potential effectiveness while underscoring the need for context-based adaptation to optimize impact across diverse humanitarian settings. To ensure cultural relevance and contextual fit for the present study, the SSAGE curriculum was adapted for use with Venezuelan migrants and Colombian returnees in Colombia through a structured human-centered design (HCD) process (Seff et al., Under Review). This participatory, multiphase approach engaged adolescent girls, male siblings, caregivers and local stakeholders in participatory activities and focus group discussions, co-design workshops, pilot sessions and expert consultations. Key adaptations included broadening eligibility, revising curriculum language, adding content on boys' mental health risks and emphasizing sibling engagement to

enhance acceptability while preserving its core gender-transformative framework (Seff et al., Under Review).

This manuscript presents endline findings from a type-1 hybrid effectiveness-implementation pilot evaluation of the SSAGE intervention, implemented with Venezuelan migrant and Colombian returnee families in Colombia. A type-1 hybrid effectiveness-implementation evaluation focuses primarily on the intervention outcomes and secondarily on understanding the implementability and implementation context (Landes et al., 2020). This design was selected because SSAGE had demonstrated feasibility and acceptability in previous humanitarian contexts, but had not been implemented with Venezuelan migrants or Colombian returnees in urban Colombia. This design allowed us to gather preliminary data on potential effectiveness while simultaneously examining whether and how the intervention could be feasibly delivered in this new setting – information essential for informing future scale-up efforts. The dual focus was particularly important given the contextual differences between camp-based settings in previous implementations and the urban, economically precarious environment faced by displaced families in Colombia. We assess the potential impact on key mental health indicators, including anxiety, depression and self-esteem, alongside key implementation outcomes, including acceptability, feasibility and barriers and facilitators to program engagement.

## Methods

### Intervention processes

SSAGE program sessions were delivered by trained community facilitators ("mentors"), who were supported by program supervisory staff responsible for accompaniment, supervision and monitoring of session delivery. Mentors were selected by the Mercy Corps team based on their knowledge of psychosocial support and mental health, as well as prior experience facilitating community-based group processes. The majority of mentors were selected from Mercy Corps staff, and, therefore, also had experience working with migrant populations. Mentors participated in a structured Training of Trainers (ToT) process before implementation. The ToT was delivered to supervisory staff and was designed to strengthen facilitators' knowledge, skills and commitment to delivering the intervention as intended, including core content related to gender norms, power and gender-based violence, as well as gender-equitable communication and caregiving skills. Training also emphasized participatory, experiential learning approaches, in which facilitators guide participants through structured activities that promote reflection, dialog and collective problem-solving rooted in lived experience.

Fidelity monitoring was conducted throughout implementation using structured tools and supervision processes. Supervisory staff conducted session observations using a structured monitoring tool, followed by an immediate debrief with the facilitator to reflect on delivery and identify areas for improvement. Ongoing implementation support also included periodic review meetings to share experiences and address emerging implementation challenges, informed by routine review of monitoring data.

### Participants and procedures

Participants for this study were recruited by Mercy Corps staff from a pool of families within the Mercy Corps database. While the SSAGE program itself is family-based, only adolescent girls were included in the effectiveness evaluation as a result of budgetary constraints (*i.e.*, effectiveness outcomes were only collected for adolescent girls). Eligibility criteria included adolescent girls who had migrated from Venezuela or who were born in Colombia and had previously emigrated to Venezuela for at least 5 years and recently returned to Colombia. To minimize community tensions between migrant and host populations, a few adolescent girls from internally displaced or host families who met all other inclusion criteria were included in the study. Other eligibility criteria included being 13–19 years old, being available, along with their family members, to participate in the SSAGE intervention for 12 weeks, and being available for endline data collection immediately following the completion of SSAGE. Original eligibility criteria dictated that adolescent girls also live with an adult male caregiver who would participate in the intervention. This requirement was removed after in-depth discussions with Mercy Corps and other local stakeholders around typical household compositions in the study context and the preference to not exclude adolescent girls who did not happen to live with a male caregiver (Seff *et al.*, Under Review). Only one adolescent girl per family was selected to participate in SSAGE and enroll in the study.

Consent and enrollment processes were carried out by the data collection team, who were trained at baseline and again at endline on ethical research processes. All data collectors were female to maximize adolescent girls' comfort during data collection. Data collectors obtained informed consent from caregivers before obtaining informed assent from adolescent girls under age 18 years; adolescent girls ages 18 and 19 years provided informed consent directly. Data collectors explained the study purpose, voluntary participation and safeguarding procedures, noting confidentiality would only be broken to connect participants at risk of harm or disclosing violence to services *via* Mercy Corps' referral pathway. A social worker trained in psychological first aid was also on call to provide remote support when a respondent expressed distress during an interview.

Ultimately, 186 adolescent girls were enrolled in the study. Additional details on sample size calculations have been published elsewhere, though it is important to note that this pilot trial is not powered to detect effectiveness (Seff et al., 2024). A random number generator was used to randomize 93 girls each to the treatment or control arm in a 1:1 ratio. Randomization was carried out by the research team and was not blinded. All 186 adolescent girls enrolled in the study completed a baseline survey questionnaire, and adolescent girls who were reached at endline completed the same survey within 2 weeks of the intervention's completion. All survey questionnaires were administered using computer-assisted personal interview software.

At endline, we also recruited six male caregivers, six female caregivers and six adolescent boys – who had participated in the SSAGE program – to enroll in the study and participate in in-depth interviews covering topics related to program implementation. Interviews were also conducted with six adolescent girls who were already enrolled in the study as part of the effectiveness evaluation. In-depth interview participants were purposefully selected to reflect a range of attendance levels. Key informant interviews were also carried out with program mentors. Informed consent was obtained for all mentors and caregivers, and both parental consent and informed assent were obtained for the adolescent boys under 18 years of age. Data collection participants were given transportation stipends to cover the costs associated with traveling to and from data collection sites.

All research activities were carried out in Spanish. All study procedures received ethical approval from the Institutional Review Board at the University of Los Andes and Washington University in St. Louis (IRB #202407174). The trial was registered with Clinical-Trials as # NCT06078124.

### Outcomes

#### Effectiveness outcomes

The primary outcomes measured in the survey questionnaire included mental illness, including anxiety and depression. Secondary outcomes included self-esteem and family attachment.

Risk of various mental illnesses was measured using the Diagnostic and Statistical Manual of Mental Disorders, Fifth Edition (DSM-5) cross-cutting youth (Narrow et al., 2013). Respondents were asked to indicate the frequency of 19 symptoms in the previous 2 weeks, selecting from the following response options: 0-Never; 1-Rarely (1 or 2 days); 2-Slightly (between 3 and 6 days); 3-Moderately (more than 7 days of the last 2 weeks); and 4-Severely (Every day in the last 2 weeks). Example symptoms included "being worried about your health or getting sick," and "feeling nervous, anxious or scared." Respondents also reported whether they had engaged in each of five behaviors related to drug and alcohol use, self-harm and suicide ideation (answering yes or no) in the last 2 weeks and whether they had ever attempted suicide in their lifetime (yes or no). The included symptoms are meant to represent 11 domains of mental disorders: somatic symptoms, sleep problems, inattention, depression, anger/irritability, mania, anxiety, psychosis, repetitive thoughts and behaviors, substance use and suicide ideation/attempts. Generally, a respondent is flagged as requiring further inquiry for a given domain if they answered '2' or higher on at least one of the symptoms in each respective domain. As such, this outcome is operationalized as 11 dichotomous variables reflecting the 11 domains.

Depression and anxiety symptomology were measured using the Revised Children's Anxiety and Depression Scale (RCADS-25) (Ebesutani et al., 2017). The RCADS-25 asks respondents to indicate how frequently they have experienced each of 25 symptoms. Example symptoms include "feeling sad or empty," and "being afraid of having to be home alone at night." Possible responses include 0-Never; 1-Sometimes; 2-Frequently; and 3-Always. An overall RCADS-25 score is created by adding the responses across all 25 items and looking up the corresponding T-score from a previously created T-score chart based on gender and grade.[1] Ages, instead of grades, were used for the transformations, whereby 7th and 8th grade correspond to 13- and 14-year-olds, 9th and 10th grade correspond to 15- and 16-year-olds, and 11th and 12th grade reflect 17–19-year-olds. The RCADS-25 showed high internal consistency at baseline (Cronbach's alpha of 0.88).

Finally, self-esteem was captured using the Rosenberg Self-Esteem Scale (RSES) (Rosenberg, 1965). The RSES asks respondents to indicate their level of agreement with 10 statements on a 4-point Likert scale ranging from "totally disagree" to "totally agree." Example statements include: "I feel that I have many good qualities," and "I don't have much to be proud of." After reverse scoring 5 of the 10 items, the final score represents the sum of all items. Scores may assume a value from 10 to 40, where higher values reflect greater self-esteem. The RSES exhibited a high internal consistency with a Cronbach's alpha of 0.77 at baseline.

A modified version of the Family Attachment Changeability Index 8 (McCubbin et al., 1996) was also used to measure family attachment and changeability. However, this measure demonstrated poor reliability (Cronbach's alpha of 0.63) and validity (as per observation and participant understanding of items) and was not included in the present evaluation.

These outcome measures were selected because they have been previously used in humanitarian and low-resource settings and have demonstrated acceptable psychometric properties across diverse cultural contexts. The DSM-5 cross-cutting tool was chosen for its ability to screen across multiple mental health domains, while the RCADS-25 and RSES are brief, validated measures that minimize participant burden. However, it is important to acknowledge that these instruments were not specifically validated among Venezuelan migrants or Colombian returnees in Colombia. Additionally, the economic precarity faced by the majority of participants in this study may have influenced their mental health symptom reporting in complex ways. Many participants faced severe material hardship, including food insecurity, housing instability and chronic unemployment. These structural stressors may be both causes and consequences of psychological distress, potentially complicating the interpretation of symptom-based measures. Future research should consider incorporating measures that more directly assess economic well-being and material hardship alongside mental health outcomes, and should prioritize local validation of mental health instruments with displaced populations.

#### Implementation outcomes

Program acceptability and feasibility were measured using the corresponding sub-scales from the Mental Health Implementation Science Tools (mhIST). The mhIST was specifically designed for use in LMICs and has demonstrated strong psychometric properties across multiple humanitarian contexts, making it particularly appropriate for assessing implementation outcomes in this study (Aldridge et al., 2022). These two subscales were administered at endline to all participants assigned to the treatment arm (Aldridge et al., 2022). The acceptability scale included 15 questions capturing several dimensions of acceptability, including whether the participant liked the program overall, liked attending, was satisfied and enjoyed learning, among others. For each item, respondents were asked to select a response from 1-A little, 2-A moderate amount and 3-A lot. Thirteen of the original 15 feasibility items were included in this evaluation (two items did not apply to SSAGE) and captured the following dimensions, among others: attendance, flexibility, timeliness, time spent and transportation money. Respondents were asked to answer yes or no (1 or 0). Items were analyzed at the individual level.

Semi-structured, in-depth interviews were also conducted with 24 program participants (including 6 adolescent girls, 6 adolescent boys, 6 female caregivers and 6 male caregivers) and program mentors to capture implementation outcomes, barriers and facilitators. All interviews were carried out in Spanish by one of two female data collectors who were not involved in program delivery and had no prior relationships with participants. Interviews with program participants asked respondents about their satisfaction, preferences, barriers, lessons learned and recommendations for the program. Similar topics were covered in the mentor interviews. All interviews were audio-recorded and transcribed into Spanish.

---

[1] https://rcads.ucla.edu/sites/default/files/2023-11/TScoreConversionTablesRCADS25YandCG.pdf.

### Data analysis

#### Effectiveness evaluation

Descriptive statistics were estimated for baseline characteristics and outcomes, separately, for the control and treatment arms. Balance at baseline between the two groups for key characteristics was assessed using *T*-tests. To examine the potential effectiveness of SSAGE in improving the outcomes of interest, we initially planned to conduct both intent-to-treat and per-protocol analyses. For the intent-to-treat analysis, linear regression models were employed to estimate the effect of the intervention on each outcome at endline. In addition to unadjusted models, we ran logistic regressions controlling for basic characteristics. A girl was defined as adhering to protocol if both she and all of her participating family members attended at least 50% of sessions (seven or more sessions). As a result of several barriers to attendance – which we explore in-depth in later sections – only 10% of girls assigned to the treatment arm were considered to have adhered to protocol. As a result, we did not conduct per-protocol analysis. Imputation was not conducted as there was no item-level missingness on outcome measures at baseline or endline.

#### Implementation evaluation

Descriptive statistics were estimated for attendance both for each participant type and at the family level. Descriptive statistics for the two mhIST subscale items were also estimated to understand key barriers and facilitators to acceptability and feasibility for adolescent girls.

Qualitative interviews were audio-recorded, transcribed and analyzed in Spanish. Before reading the transcripts, three research team members created a codebook deductively, based on previously published definitions and conceptions of implementation outcomes (Proctor et al., 2011), with a focus on acceptability, feasibility and fidelity. Following coding of all transcripts with this codebook, each team member read the transcripts again, this time carrying out inductive open coding, driven by the data itself. To minimize bias, coding was conducted by multiple researchers, with regular analytic meetings to discuss discrepancies and refine interpretations. All coding took place using the Dedoose software ('Dedoose', 2024). Finally, applied thematic analysis was employed, whereby the team held meetings to discuss relationships between deductive and inductive codes to identify themes and organize findings.

### Results

In August 2024, 186 adolescent girls were enrolled in the study (Figure 1) and completed the baseline survey. Ninety-three girls each were then randomly assigned to either participate in SSAGE with their families or serve as the control group. Of these, 158 (85%) adolescent girls completed the endline survey in December 2024, with equal loss-to-follow-up in both the treatment and control groups.

#### Baseline characteristics

Baseline characteristics are presented in Table 1. On average, girl participants were 15.83 years old, and 82.3% were currently in school. Approximately 82.3% of adolescent girl participants were refugees or migrants, with 1.1%, 12.4% and 5.3% being internally displaced, returned Colombians or part of the Colombian host population, respectively. There was substantial variation in the percentage of participants who were flagged as requiring further inquiry for the DSM-5 domains. Exhibiting repetitive thoughts (73.1%), inattentiveness (72.0%), somatic symptoms (65.1%) and depressive symptoms (65.6%) were the most commonly flagged domains, while substance use (21.0%) was the least commonly cited. In the full sample, the average RCADS-25 t-score was found to be 54.03, and the average RSES was 29.21. All characteristics and outcomes were balanced between treatment and control groups at baseline except for household size, with participants in the control group reporting living with 6.04 people, including themselves, as compared to 5.35 in the treatment group (*P* = 0.012).

Additionally, no outcomes of interest were found to be associated with loss-to-follow-up.

#### Program effectiveness

Regression analysis revealed no treatment effects on any of the DSM-5 categories, the RCADS-25 or RSES, regardless of whether or not models controlled for basic characteristics (see Table 2).

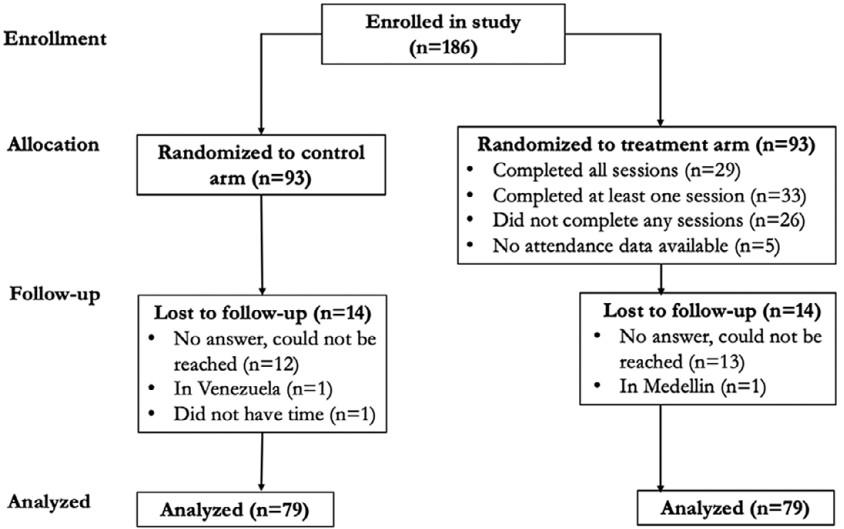

**Figure 1.** CONSORT Flow Diagram.

**Table 1.** Baseline characteristics

| | Full sample (*n* = 186) | Control (*n* = 93) | Treatment (*n* = 93) | *P*-value |
|---|---|---|---|---|
| Age | 15.83 [1.87] | 15.82 [1.79] | 15.84 [1.96] | 0.938 |
| Currently in school | 153 (82.3%) | 76 (81.2%) | 77 (82.8%) | 0.849 |
| Household size | 5.70 [1.99] | 6.04 [2.14] | 5.35 [1.77] | 0.012 |
| Status | | | | 0.741 |
| Refugees or migrants | 153 (82.3%) | 79 (85.0%) | 74 (79.6%) | |
| Internally displaced | 2 (1.1%) | 1 (1.1%) | 1 (1.1%) | |
| Returned Colombians | 23 (12.4%) | 9 (9.7%) | 14 (15.1%) | |
| Host population - Colombians | 8 (4.3%) | 4 (4.3%) | 4 (4.3%) | |
| DSM categories | | | | |
| Somatic | 121 (65.1%) | 64 (68.8%) | 57 (61.3%) | 0.284 |
| Sleep | 80 (43.0%) | 38 (40.9%) | 42 (45.2%) | 0.556 |
| Inattentiveness | 134 (72.0%) | 66 (71.0%) | 68 (73.1%) | 0.746 |
| Depression | 122 (65.6%) | 65 (69.9.%) | 57 (61.3%) | 0.219 |
| Anger/Irritability | 107 (57.5%) | 57 (61.3%) | 50 (53.8%) | 0.302 |
| Mania | 65 (34.9%) | 30 (32.3%) | 35 (37.6%) | 0.445 |
| Anxiety | 117 (62.9%) | 58 (62.4%) | 59 (63.4%) | 0.880 |
| Psychosis | 70 (37.6%) | 36 (38.7%) | 34 (36.6%) | 0.764 |
| Repetitive thoughts | 136 (73.1%) | 71 (76.3%) | 65 (69.9%) | 0.324 |
| Substance abuse | 39 (21.0%) | 19 (20.4%) | 20 (21.5%) | 0.858 |
| Suicidality | 76 (40.9%) | 35 (37.6%) | 41 (44.1%) | 0.374 |
| Revised Child and Anxiety Depression Scale (RCADS–25) t-score | 54.03 [14.49] | 54.31 [13.50] | 53.76 [15.49] | 0.797 |
| Rosenberg Self-Esteem Score (RSES) (10–40) | 29.21 [4.19] | 29.17 [4.11] | 29.25 [4.28] | 0.903 |

*Note:* Statistics are mean [SD] or *n* (%). *P*-values correspond to *t*-tests or chi-squared tests of differences between the treatment and control groups.

Null findings are consistent with the pilot study design, which was not powered to detect treatment effects and focused primarily on implementation outcomes.

### Implementation evaluation

#### Attendance
Attendance data were available for 88 of the 93 families assigned to the treatment arm (the remaining five families were lost to follow-up following baseline data collection and thus did not engage in the program). Among these 88 families, there were 88 adolescent girl participants, 82 adolescent boys, 83 female caregivers and 45 male caregivers. For those participating, attendance was generally low, with girls, boys, female caregivers and male caregivers attending an average of 6.7, 5.0, 4.7 and 4.5 sessions, respectively (Figure 2). Protocol adherence, defined as attending at least 50% of sessions, was also low, particularly for non-adolescent girl participants. Approximately 57%, 40%, 39% and 38% of adolescent girls, boys, female caregivers and male caregivers, respectively, attended at least 50% of sessions.

Analysis of the mhIST data on acceptability and feasibility, as well as the in-depth interviews, helps to contextualize these attendance findings and highlight key barriers and potential facilitators to attendance.

#### Acceptability
Despite low overall attendance, the program was well-received by those who did attend. Quantitative data from the mhIST demonstrated

strong endorsement of the intervention's relevance and content. On a 4-point Likert scale, adolescent girls gave high scores for general program satisfaction (3.80), enjoyment (3.93) and the usefulness of the skills learned (3.89) (Table 3). Girls also reported high levels of satisfaction with their mentors (3.82) and confidence in their mentors' abilities (3.87).

These findings were strongly echoed in the qualitative interviews, where participants across all groups described SSAGE as a safe, affirming space that fostered meaningful reflection, interpersonal connection and critical dialog. Adolescents described the program as an opportunity to challenge dominant social norms, engage with unfamiliar but relevant material and gain knowledge that helped them better understand themselves and their relationships. As one adolescent boy reflected, "*It gave us the chance to think freely and share ideas that society usually puts in our heads about how we're supposed to be or act*" (IDI 7, AB). Similarly, an adolescent girl participant appreciated that the curriculum prompted reflection on fairness and equity: "*We talked about the different kinds of violence… and how it's important to treat everyone fairly and not to discriminate against people…*" (IDI 10, AG).

Caregivers also reported increased awareness of family dynamics and gender roles. Female caregivers in particular described SSAGE as eye-opening, prompting reflection on long-standing norms around domestic labor and power in the household. As one woman explained, "*Women are often responsible for managing the household. Men can and should help… Why must women do it all? That is something I learned*" (IDI 3, FC). For male caregivers, the

**Table 2.** Treatment effects

| | Unadjusted | Adjusted |
|---|---|---|
| | B [95% CI] | B [95% CI] |
| DSM categories | | |
| Somatic | 0.025 | 0.031 |
| | [−0.13,0.18] | [−0.13,0.19] |
| Sleep | −0.051 | −0.049 |
| | [−0.20,0.10] | [−0.19,0.09] |
| Inattentiveness | 0.114 | 0.103 |
| | [−0.04,0.27] | [−0.05,0.26] |
| Depression | −0.051 | −0.061 |
| | [−0.21,0.11] | [−0.22,0.10] |
| Anger/Irritability | −0.025 | −0.043 |
| | [−0.18,0.13] | [−0.20,0.12] |
| Mania | 0.114 | 0.108 |
| | [−0.03,0.26] | [−0.05,0.26] |
| Anxiety | −0.025 | −0.029 |
| | [−0.18,0.13] | [−0.19,0.13] |
| Psychosis | 0.051 | 0.059 |
| | [−0.09,0.19] | [−0.08,0.20] |
| Repetitive thoughts | 0.127 | 0.109 |
| | [−0.02,0.28] | [−0.04,0.26] |
| Substance abuse | 0.051 | 0.064 |
| | [−0.08,0.19] | [−0.07,0.19] |
| Suicidality | 0.013 | 0.004 |
| | [−0.14,0.17] | [−0.15,0.16] |
| Revised Child and Anxiety Depression Scale (RCADS–25) *t*-score | −0.468 | −0.562 |
| | [−5.83,4.90] | [−5.91,4.78] |
| Rosenberg Self-Esteem Score (RSES) (10–40) | 0.215 | 0.272 |
| | [−0.95,1.38] | [−0.89,1.43] |

*Note*: Adjusted models control for age, whether or not the respondent is currently in school, refugee status and household size. Beta coefficients are statistically significant at \*P < 0.01, \*\*P < 0.05 and \*\*\*P < 0.001.

opportunity to explore fatherhood, emotional expression and marital relationships in a supportive setting stood out. One man shared how the sessions encouraged him to reexamine his values: "*We discussed our childhood, adolescence, and our roles as spouses. We talked about how we care for our children, our wives, and our families... It's always helpful to revisit and be motivated by these discussions*" (IDI 15, MC).

Mentors also offered insight into the elements of the program that contributed to its perceived acceptability. They emphasized the importance of building horizontal, nonjudgmental relationships with participants, creating an atmosphere in which adolescents and caregivers felt safe to speak openly. As explained by one male mentor: "*I always tried to start from a comfortable place. Kids would say, 'Teacher, I don't picture you as a teacher...' I'd tell them, 'No worries! The main thing is that you feel comfortable opening up to me'*" (IDI 17, MM). One female mentor felt that the safe space and mentor provided a sense of ease for the adolescent girls. She noted,

"*I think what they really enjoyed was having a place to just talk. They could share stuff they probably wouldn't feel comfortable bringing up with their mom...*" (IDI 19, FM).

### Feasibility

Quantitative data from the feasibility subscale of the mhIST suggest that barriers to feasibility may better explain the low levels of attendance. Only 58% of girls said they could attend weekly sessions without difficulty, and just 52% found it easy to get away from daily responsibilities (Table 3). Constraints related to finances and resources were the most cited barriers to attendance. Less than half (44%) of adolescent girls had enough money to pay for transportation to and from sessions, and only 34% reported having enough money for other participation-related needs.

Qualitative interviews reinforced these quantitative findings from adolescent girls, painting a fuller picture of the daily constraints that all participants faced. Families described living "day to day," often relying on informal or short-term employment that prevented flexibility to attend multi-week programming. Attending SSAGE often meant sacrificing wages or missing work entirely – a tradeoff that many households, particularly those already facing extreme vulnerability, could not afford. As one male caregiver explained, "*Some of my friends couldn't make it to the program because they had to work. They often said things like, 'I can't come; it's a workday, and I need to earn some money.' A lot of us are living day by day... It would be awesome if the program could help out financially. This way, people wouldn't have to choose between attending and making money*" (IDI 15, MC).

Mentors echoed these concerns, noting that the program's transportation stipend was insufficient to meaningfully offset participants' opportunity costs. The stipend provided – ~$2.75 – fell short of what participants typically earned in a single day, which mentors estimated to be closer to $7. One mentor emphasized, "*If the incentives were increased slightly, it could lead to higher attendance... While attending these sessions is beneficial, it's also unfair for individuals to lose the occasional income they depend on*" (IDI 20, MM). Venue accessibility and related travel costs also affected program feasibility. Although sessions were held in locations intended to be close to the communities, some participants still faced long commutes between home, work and the program site. In certain cases, transportation challenges were exacerbated by seasonal flooding, poor infrastructure or safety concerns in peripheral neighborhoods. One male caregiver described his experience: "*Where did we have the meetings? It was quite far away from where I was working... If I took the bus, it took longer, but if I went by motorbike, I could get there much quicker. However, that option was more expensive in terms of transport*" (IDI 8, MC). A female mentor elaborating on the tension between distance and safety shared: "*I believe the ideal approach is to be as close to the community as possible. However, this also presents challenges. While we provided a transportation subsidy to help access those areas, it ultimately wasn't effective because no motorcycle drivers were willing to go there*" (IDI 19, FM).

Beyond structural constraints, mentors also raised important concerns about certain participants' emotional readiness. While the SSAGE program was designed to promote mental health for adolescent girls and their families, many participants – particularly caregivers – entered the program in acute states of emotional distress that sometimes exceeded what the intervention was equipped to address. This distress was often rooted in prior experiences of violence, trauma or the compounding stresses of displacement and poverty. One mentor recalled a session early in

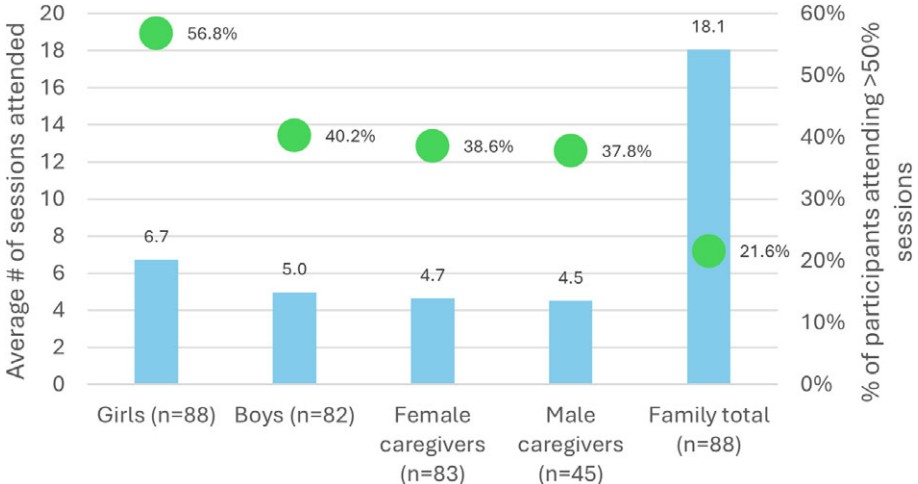

**Figure 2.** Program attendance by participant type and whole family.

the program: "*During one of the sessions, all the caregivers began to cry. I thought, 'Oh my God, what should I do?' They were clearly overwhelmed… some had experienced violence, while others were burdened by their migration experiences*" (IDI 21, FM).

In response, mentors described providing psychological first aid or conducting impromptu one-on-one sessions before being able to deliver the core curriculum. They emphasized that this emotional overload was not caused by the content of SSAGE itself but rather reflected participants' unaddressed psychological needs. As one mentor explained: "*Many caregivers began the program feeling emotionally overwhelmed… I found it necessary to provide psychological first aid and, in some cases, individual sessions before delivering the main content*" (IDI 21, FM). These insights point to an important consideration for future implementation: programs like SSAGE may be more effective if paired with early assessments of psychological well-being and a front-loaded component that offers therapeutic support to participants with high emotional needs. Doing so could strengthen participants' capacity to engage fully with the program and help ensure that content is received as intended.

*Fidelity*

Although mentors demonstrated a strong understanding of SSAGE's goals – namely, to prevent adolescent girls' mental illness through a whole-family, gender-transformative approach – they also felt that barriers related to scheduling and time constraints may have compromised the program's fidelity during implementation. These challenges affected both the consistency and depth with which the curriculum was delivered, raising concerns about the degree to which the program was implemented as originally intended.

In some cases, sessions had to be canceled due to inclement weather, venue inaccessibility or poor attendance. As a result, mentors reported that they were sometimes forced to combine two or even three sessions into a single 2-h meeting. While this approach allowed them to cover the full curriculum within the available timeline, it compromised the quality of delivery and led to discussions that were "quite brief and lacked depth." Participants were presented with large volumes of material in a short period, which limited the opportunity for discussion, reflection and participatory learning – core principles of the SSAGE model. As one female mentor explained: "*I had to rearrange some activities or plan them differently to prevent people from getting bored… particularly*

when we had to facilitate one or multiple sessions in a single meeting. While it wasn't so demanding for me, it could be overwhelming for the participants because they were being presented with a lot of information at once*" (IDI 19, FM).

Mentors also attempted to deliver virtual makeup sessions. These sessions provided an alternative mode of engagement, but mentors uniformly agreed that virtual delivery was not a sufficient substitute for in-person interactions – especially given the sensitive, relational content of the program. One mentor described the limitations of remote delivery: "*I held virtual meetings and made phone calls. But for me, face-to-face interaction is really very important. It's not the same to intervene when you can be seen, heard and felt… especially given the issues we are addressing. I think that while virtual sessions can be an alternative, they shouldn't be the only option. That wouldn't make sense.*" (IDI 21, FM). These adaptations – though often necessary – may have undermined core components of the SSAGE methodology, which relies heavily on building group cohesion and facilitating open discussion. The inability to maintain session consistency and the shift toward compressed or virtual delivery formats likely diminished the opportunity for participants to fully absorb and internalize program content.

Taken together, the quantitative and qualitative findings suggest that while SSAGE was highly valued by participants, its feasibility was significantly constrained by the economic precarity, logistical barriers and emotional burdens many families faced. Addressing these challenges directly will be critical for improving attendance and scaling and sustaining future implementation.

**Discussion**

This hybrid type-1 effectiveness-implementation study presents findings from a mixed-methods randomized controlled trial (RCT) of SSAGE, a gender-transformative, whole-family intervention designed to improve mental health among forcibly displaced adolescent girls. Findings around effectiveness revealed no detectable treatment effects on adolescent girls' mental health or self-esteem outcomes, aligning with pilot RCT methodology aiming to assess implementation feasibility and generate preliminary effect size estimates. While the lack of statistically significant treatment effects may be due to the smaller sample size that is characteristic of pilot trials, these null findings were likely also influenced by low attendance and implementation challenges. Among participants

**Table 3.** mhIST responses: Acceptability and feasibility items

| Acceptability | Average score range: 1–4 |
|---|---|
| Overall, did you like SSAGE? | 3.80 |
| Did you like attending SSAGE sessions? | 3.93 |
| Did you feel satisfied with the SSAGE services received? | 3.92 |
| Did you enjoy learning SSAGE? | 3.93 |
| Do you feel that the skills you learned in SSAGE are useful? | 3.89 |
| Do you feel that the components of SSAGE make sense to you? | 3.80 |
| Did you feel comfortable raising questions to your mentor? | 3.75 |
| Did you feel that your mentor listened to your concerns and questions about SSAGE? | 3.73 |
| Did you feel satisfied with your mentor's abilities in SSAGE? | 3.82 |
| Did you feel that your mentor addressed any questions or concerns you had about SSAGE? | 3.63 |
| Did your mentor take an interest in you? (UKV) | 3.93 |
| Was your mentor available when you needed to talk to him/her? | 3.73 |
| Did you feel that you could trust your mentor? | 3.69 |
| Did you feel that your mentor was qualified enough to deliver SSAGE? (UK, UZ) | 3.87 |
| Did you feel that you understood the way things were explained to you during SSAGE? | 3.82 |
| **Feasibility** | **% responding "yes"** |
| Have you been able to attend the weekly sessions of SSAGE without difficulty? | 58.23% |
| Were SSAGE sessions scheduled with enough flexibility to meet your needs? | 83.54% |
| Was your mentor on time for your sessions? | 86.08% |
| Was it easy for you to get away from your responsibilities (e.g., schooling, housework) to attend SSAGE? | 51.90% |
| Was the amount of time you spent doing SSAGE activities at home each week manageable? | 82.28% |
| Did you have enough money to pay for transport to get to SSAGE sessions? | 44.30% |
| Did you have enough money to pay for any other things you needed to get to the SSAGE sessions? | 34.18% |
| Did you have enough resources (phone, talk time) to communicate with your mentor when needed? | 67.09% |
| Did you have the emotional support that you needed from your family and friends to attend SSAGE? | 87.34% |
| In general, did you feel safe to travel to weekly SSAGE sessions? | 86.08% |
| Did you feel the place you met with your mentor and peers was safe? | 79.75% |
| Did you feel the place you met with your mentor and peers was private and confidential? | 81.01% |
| Do you believe people in your community could participate in SSAGE without fear of how others would view them? | 88.61% |

who engaged with the program, acceptability was high, especially relating to the program content and mentor relationships. Nonetheless, a range of barriers – including economic precarity, household member responsibilities and difficulties accessing transportation and venues – limited feasibility. These feasibility constraints, in turn, led to session rescheduling, compression and virtual delivery, which collectively compromised the program's ability to deliver core content as intended.

Despite these delivery challenges, both quantitative and qualitative implementation data point to the program's potential. Participants consistently expressed appreciation for SSAGE as a space to explore complex emotions, challenge social norms and strengthen interpersonal relationships. The program's relevance and resonance were clear across age and gender. Adolescents valued the opportunity to reflect on gender, power and emotional health,

while caregivers – especially men – highlighted new insights related to parenting, gender roles and family connection. This broad-based acceptance suggests that SSAGE not only filled an important psychosocial gap but may have paved the way for longer-term changes in family norms and adolescent well-being. Qualitative findings also suggest that family members may be experiencing early shifts in attitudes and behaviors aligned with SSAGE's theory of change (Seff et al., 2024) – even if those shifts have not yet translated into measurable improvements in adolescent girls' mental health. These early changes may represent important precursors to improved outcomes over time. Future evaluations of SSAGE and similar programs should consider tracking intermediate outcomes among all family members, in addition to primary outcomes for adolescent girls.

The evaluation also uncovered a clear and urgent need for mental health support among participants – underscoring the

importance of pairing preventive interventions like SSAGE with more robust psychosocial services when appropriate. Several participants began the program in acute emotional distress, in some cases to a degree that interfered with being able to fully engage with the curriculum. This finding aligns with prior research documenting elevated rates of mental illness and suicide among Venezuelan migrants in Colombia, with high levels of post-traumatic stress attributed to extreme instability in their country of origin, loss of family, homes and other social connections, risks to safety on the migration journey, precarious legal statuses in Colombia and challenges in securing income-generating opportunities, among others (Espinel et al., 2020; Alarcon et al., 2022). While adolescent girls who reported suicidality during baseline data collection were referred to additional mental health services, not collecting data from other family members meant they were not screened in the same way. This is a critical gap, as caregiver mental health is a well-established predictor of parenting practices and adolescent well-being (Murphy et al., 2017; Meinhart et al., 2023). Future program designs should consider universal screening for all family members, with referral mechanisms in place for those in crisis. Doing so would not only support those most in need but also strengthen families' capacity to engage with and benefit from interventions like SSAGE.

However, for SSAGE – and other whole-family interventions in similar settings – to realize their full potential, future implementation approaches must include explicit strategies to improve attendance. Participants faced a range of barriers to sustained program engagement, including inflexible work schedules, limited financial resources, inaccessible or unsafe venues and limited affordable transportation options. These realities reflect the chronic precarity faced by many forcibly displaced Venezuelan families in urban Colombia. These findings echo patterns documented in other family-based or mental health interventions in low-resource and humanitarian settings, where attendance is often constrained by competing demands on time and limited financial flexibility (Puffer et al., 2016; Shenderovich et al., 2018; Peycheva et al., 2023). While previous studies have identified participant characteristics associated with low attendance (Eloff et al., 2014; Janowski, 2020), there is still limited evidence on concrete, effective strategies to improve engagement in complex, resource-constrained environments.

Based on our findings, several adaptations may improve participation in this context, including increasing the size of monetary incentives, adding sessions on income-generating activities, offering more flexible scheduling options and delivering sessions in or near locations that align with participants' daily routines. These considerations and potential implementation strategies should be addressed during program design and co-creation – not only identified in hindsight. Unfortunately, because attendance was not an issue in the study team's previous implementations of SSAGE in other settings, attendance strategies were not explicitly included as a point of discussion during this pilot's HCD phase. As a result, feasibility issues in Colombia emerged only during implementation, undermining both attendance and fidelity. While the Colombia implementation was carried out in urban areas, previous implementations took place in rural and camp settings within Nigeria, Jordan and Niger. Differences in livelihoods, opportunity costs, cost of living and transportation between these two contexts may have also affected differing rates of attendance.

Strategies to increase attendance should not focus solely on incentivizing attendance without also understanding and addressing participants' basic needs and preferences; changes to the intervention itself, such as integrating opportunities to bolster livelihoods, as noted above, may increase both participants' capacity and intrinsic motivation to participate in the intervention. A more anticipatory and participatory approach to feasibility – including localized mapping of opportunity costs and structural barriers – is thus especially important when adapting interventions across humanitarian settings. Importantly, implementation strategies that are shaped by community priorities and support families in meeting their basic needs may have the added benefit of freeing up the cognitive bandwidth needed to engage in discussions on gender norms, power relations and other sensitive topics. Implementation science research is also needed to rigorously test the effectiveness and cost-effectiveness of different incentive models for improving program participation. Without improvements in attendance, interventions like SSAGE may struggle to achieve intended outcomes regardless of their acceptability or theoretical grounding.

Findings from this study should be considered alongside a few limitations. First, the low program attendance limits our ability to draw conclusions about SSAGE's effectiveness. While a pilot study is generally not powered to detect impact, the lack of participation among many treatment participants and their family members further compromised our ability to detect changes in outcomes. As such, the null findings reported here should be interpreted with caution. Nonetheless, these findings now provide solid baseline estimates of key outcomes, which can in turn inform sample size calculations for future, fully powered RCTs. Second, the self-reported nature of our effectiveness, feasibility and acceptability outcomes of interest may introduce reporting bias, especially if participants felt motivated to portray themselves or the program in a positive light. Third, an important limitation concerns the cultural validity of our outcome measures. All instruments used in this study were developed in anglophone, Global North contexts and, while previously used in humanitarian settings, were not specifically validated among Venezuelan migrants or Colombian returnees in urban Colombia. The RCADS-25 and RSES, for instance, measure symptomology based on Western conceptualizations of anxiety, depression and self-esteem, which may not fully capture how psychological distress and well-being are experienced and expressed in this population. Similarly, while the DSM-5 cross-cutting tool offers broad screening across mental health domains, its symptom-based approach may not be culturally appropriate for all participants. Finally, because the program was adapted to the Colombian context through a structured HCD process, findings from the study may not be generalizable for camp-based or rural settings.

## Conclusion

SSAGE holds promise as a gender-transformative, family-based approach to improve the mental health and psychosocial well-being of displaced adolescent girls in Colombia. However, effective delivery in urban migrant settings requires careful attention to structural constraints, participants' emotional readiness and delivery modality. In this study, participants valued the program, yet economic pressures, transport and venue challenges, and schedule disruptions hurt attendance and fidelity. These conditions likely diluted measurable effects on girls' anxiety, depression and self-esteem.

The results underscore the need to prioritize implementation outcomes during the design and adaptation of gender-transformative interventions. Factors such as delivery modality and cultural relevance significantly influence program accessibility and impact. Future research should focus on assessing implementation processes,

including feasibility and participant engagement, to strengthen the scalability of programs like SSAGE in humanitarian contexts. By addressing these challenges, such interventions can better support displaced adolescent girls, enhancing their resilience and mental health.

**Open peer review.** To view the open peer review materials for this article, please visit http://doi.org/10.1017/gmh.2026.10161.

**Supplementary material.** The supplementary material for this article can be found at http://doi.org/10.1017/gmh.2026.10161.

**Data availability statement.** Other researchers may apply to be granted access to the data for this study, following NIH and Washington University institutional guidelines. Researchers may send such requests to the PIs of the study (LS and IS). Each request will be evaluated on a case-by-case basis. A confidentiality statement and data sharing agreement may be required before any data is shared.

**Acknowledgments.** The authors would like to acknowledge all the participant families who volunteered their time for this pilot study and the SSAGE program.

**Author contribution.** Conceptualization: L.S., I.S., A.H.R. and J.D.; Data curation: I.S., N.C., C.R. and E.T.A.P.; Formal analysis: I.S., N.C. and A.H.R.; Funding acquisition: L.S. and I.S.; Methodology: I.S., A.H.R., L.S. and B.P.; Writing (original draft preparation): I.S., J.D. and F.G.; Writing (review and editing): All authors.

**Financial support.** This research was funded by the National Institute of Health (R34MH134078).

**Disclosure statement.** The authors report there are no competing interests to declare.

**Ethics statement.** All study procedures received ethical approval from the Institutional Review Board at the University of Los Andes and Washington University in St. Louis.

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
