## [Reviewer Report]

Thank you very much for allowing me to review the manuscript “Feasibility, Acceptability, and Implementation of a Whole-Family Mental Health Intervention for Displaced Adolescent Girls in Colombia: A Mixed-Methods Pilot Randomized Controlled Trial.” I think this addresses a globally relevant topic and is in line with the scope of the Journal. Below, I will mention some limitations and some suggestions that could help strengthen the manuscript:

- Mention the number assigned to the randomized clinical trial and where it was obtained.

- The introduction adequately reviews displacement and the factors associated with mental health in these contexts. However, there is no mention of programs or implementation strategies in humanitarian settings that would situate the reader and help discuss the results later. I recommend a major review from implementation science in low- and middle-income countries (Alonge, O., Brooks, M.B. Implementation science derived from low- and middle-income countries is essential for advancing global health. BMC Glob. Public Health 3, 84 (2025). https://doi.org/10.1186/s44263-025-00206-1), especially to discuss how contexts and people’s voices are fundamental in the design of strategies (Chase, L., & Mosse, D. (2025). The Moral Blind Spots of Evidence-Based Psychiatry: Learning from Britain’s Trial of “Peer-Supported Open Dialogue.” Medical Anthropology, 1–14. https://doi.org/10.1080/01459740.2025.2563253)

- The design of a study Hybrid could be mentioned and defined in the methods (Landes, S. J., McBain, S. A., & Curran, G. M. (2019). An introduction to effectiveness-implementation hybrid designs. Psychiatry Research, 280, 112513. https://doi.org/10.1016/j.psychres.2019.112513).

- It could be better explained why this strategy was chosen for the study population

- CONSORT-style transparency is needed - Improve the clarity, completeness, and transparency of how randomized controlled trials (RCTs) are being conducted - recruitment flow, refusals, missing data handling, interviewer characteristics. It is also important to note other ethical considerations, especially in the process of involving migrant adolescents in a Pilot Randomized Controlled Trial.

- It is important to point out why the instruments were chosen. This choice could be better oriented to the results of the strategy itself. A limitation could be mentioned regarding the validation of instruments and the lack of addressing basic needs, which are compromised in many migrants and could bias the instruments that measure symptoms, especially when 82.3% of adolescent girl participants were refugees or migrants. I consider the use of mhIST to be a success.

- The results are well presented. They are consistent with the objective of the study.

- The discussion could be improved with the aspects mentioned previously.

I consider the study to be well written, with acceptable methods and results. However, I found it difficult to understand why this strategy was chosen for this context, and how and how it could help the people in the study, at a specific moment of psychosocial vulnerability. Perhaps understanding this could also help analyze the strategy’s viability and effectiveness.

---

## [Reviewer Report]

The manuscript addresses an important and understudied issue: the mental health of migrant adolescent girls in humanitarian contexts.

The introduction presents a strong rationale, with clear and persuasive argumentation for why this population warrants targeted attention. However, the framing would benefit from more explicit engagement with previous research on the SSAGE intervention itself. While the methods section describes the program, the introduction should outline where SSAGE has been tested in humanitarian settings, specify the geographic contexts in which it has been implemented, and summarize what is known about its cultural applicability, feasibility, and acceptability. Incorporating existing evidence on prior implementations, including lessons learned, cultural adaptations, unintended consequences, and documented benefits, would more effectively situate the present study within the broader literature.

The description of the intervention would be strengthened by providing additional detail about its structure and intensity. Although the manuscript notes that it was a 12-week program, it does not specify the intervention structure as defined in the original protocol, for instance, how many sessions were intended, the planned duration of each session, whether the program was designed for individual families or group delivery. Without this information, it is difficult to assess how closely the implementation aligned with the intended model or to contextualize the participation challenges described. Compensation procedures should also be described in the methods rather than appearing only in the qualitative results. These details are essential for interpreting participation challenges and for understanding the overall burden placed on families.

There is also a need for greater clarity regarding how the intervention was delivered. The methods section does not clearly specify whether sessions occurred online, in person, or in a hybrid format, which is particularly important given the conclusions drawn about challenges with in-person attendance. Although the results section later notes that some sessions were conducted online, this information should appear in the procedures so that readers can fully understand the implementation conditions from the outset. In addition, the fidelity section would benefit from reporting a clear breakdown of how many sessions from the protocol were delivered in person versus online, and if this aligns with the original protocol or planning for this study. It would also be useful to note whether there were differences in session duration across formats, for example, whether online and in-person sessions followed the prescribed one-hour structure (or whatever the intended session length is, which is not specified in the manuscript) or whether timing varied. Providing these details would strengthen the transparency of the implementation description and support interpretation of feasibility findings.

The manuscript also lacks information on fidelity monitoring. It is unclear what training facilitators received, who were the facilitators (undergrads, graduate students, contractors, psychologists, social workers) whether their delivery was supervised, or whether fidelity checks were conducted throughout the implementation period, there is no description of video or audio recordings from session to check fidelity. These elements are necessary to contextualize the findings, particularly when the results highlight implementation challenges. Without this information, it is difficult to determine whether observed difficulties reflect issues with the intervention, contextual barriers, or deviations from intended delivery.

For the quantitative measures, further clarification is needed. The authors should report internal consistency estimates (e.g., Cronbach’s alpha) for the RCADS-25 and RSES within this sample. It is also unclear who completed each measure. For example, the manuscript notes that “all participants” completed the mhIST, but it is not evident whether this refers only to adolescent girls or to all family members involved in the intervention. This same issue arises for the primary outcomes, which need to be clearly attributed either to adolescents alone or to the entire family unit. More broadly, the manuscript treats the adolescent girl as the primary unit of inference, yet SSAGE is explicitly designed as a family-based intervention. The authors should clarify whether any analytic adjustments were made for the nested structure of the data, such as family-level clustering, and why caregiver or sibling data were not analyzed as outcomes despite their central role in the program’s theory of change. Indicating whether mixed-effects models or other hierarchical approaches were considered would help readers assess whether the analytic strategy appropriately reflects the design and aims of a whole-family intervention. The qualitative methods require additional methodological transparency as well. The manuscript should indicate who conducted the interviews, whether interviewers were previously known to participants, and what steps were taken to minimize bias. The software used for qualitative data management and analysis should also be specified for reproducibility purposes.

The results section is generally clear, though some sentences, such as the statement asserting that the program shows potential despite delivery challenges, could be rewritten for readability and to better convey the relationship between the barriers identified and the observed outcomes.

In the discussion section, the authors note that attendance has not been an issue in other implementations of SSAGE, yet no references are provided to support this claim. Adding citations and offering a reflection on why the Colombian context may differ from previous implementations would strengthen the argument and demonstrate a deeper engagement with implementation science literature. Additionally, the introduction to the limitations section is awkwardly phrased; a more direct opening would improve readability.

The discussion section appropriately highlights feasibility constraints but could more directly leverage these insights to outline actionable and forward-looking recommendations. Beyond reiterating the need for improved attendance, the manuscript would benefit from specifying strategies that the next iteration of SSAGE could test. These could include transportation supports that offset opportunity costs more meaningfully, decentralized venues aligned with families’ daily movement patterns, flexible or staggered scheduling models, or brief emotional-readiness sessions to support caregivers experiencing acute distress before the group-based curriculum begins.

Overall, this study addresses a highly relevant topic and offers valuable insights into the implementation of a mental health intervention for migrant adolescent girls. Clarifying methodological details, strengthening the contextual grounding of the intervention, and deepening the engagement with the broader literature will substantially enhance the manuscript’s clarity, rigor, and contribution to the field.

Typos:

The disparate nature of these mental health challenges necessitates: necessitate (subject is singular)

These individuals engage in parallel, sessions: remove comma

we run logistic regressions: should be past tense -ran-

---

## [Reviewer Report]

This manuscript presents a timely and well-executed mixed-methods pilot RCT evaluating SSAGE, a gender-transformative, whole-family intervention for displaced adolescent girls in Colombia. The study is clearly written and makes a meaningful contribution to the implementation science literature in humanitarian mental health. The mixed-methods design is a strength, and the findings offer valuable insights into feasibility, acceptability, and contextual barriers. Some areas would benefit from clarification and expansion. More information is needed on how mentors were selected and trained, as they played a central role in program delivery and participant experience. Briefly describing their qualifications, training in gender and mental health content, and any supervision or support they received would strengthen understanding of implementation quality and the feasibility of scaling the intervention. The discussion of participants’ emotional distress is important and could be deepened by considering screening procedures and stepped-care approaches for future implementations. It would also be helpful to acknowledge the potential for bias in acceptability ratings given the low attendance. In addition, the discussion could more explicitly address how contextual differences between Colombia and previous SSAGE sites contributed to feasibility challenges, and what adaptations (e.g., improved incentives, flexible scheduling) may support future implementation. Despite these limitations, the study provides valuable baseline data and important lessons for adapting and scaling SSAGE in humanitarian settings.

---

## [Reviewer Report]

This manuscript is well written and presents important findings. However, the manuscript requires minor revision before it is ready for publication. Key areas needing attention include clarifying the description of participant recruitment across sections, providing greater detail in the qualitative analytic approach and broadening the discussion section to incorporate more critical literature. With these revisions, the manuscript has strong potential to make a meaningful contribution to the literature on gender-transformative programming within humanitarian and displacement-affected settings.

Background

I recommend expanding your discussion of gender and resilience among internally displaced populations in Colombia by engaging with the following article, which offers relevant empirical and conceptual insights: BMJ Global Health 6(10): e005770.

Methods (Line 124, p. 4)

You refer to “participatory activities,” but it would be helpful to specify which activities were implemented. Given that “participation” has increasingly become a buzzword in both academic and intervention-design settings, providing detail about the nature and purpose of these activities would significantly strengthen the methodological rigor of the paper.

Participants & Procedures (Line 146)

You note that all data collectors were female. This is an interesting and potentially intuitive decision, but it should be explicitly justified. Every methodological choice in a research design benefits from a clear rationale, particularly when it may have influenced participants’ comfort, disclosure, or the quality of data collected.

Measures

While this may be beyond the direct scope of the article, I encourage you to reflect on the cultural adaptation of mental health and related psychosocial scales developed in anglophone or Global North contexts. When transferred to socioculturally distinct settings, such measures may have important limitations. I also wonder whether a locally validated scale assessing changes in gender-norm attitudes might have been more appropriate for evaluating the program’s effectiveness.

Clarification on Interview Recruitment (Lines 163–166 vs. Line 229)

There appears to be an inconsistency in the description of how interview participants were recruited. In the first instance, you state that six male caregivers, six female caregivers, and six adolescent boys were recruited at endline, and that adolescent girls were selected from the existing study participants (the use of “but” is confusing here). Later, you describe interviews with adolescents and caregivers more generally as part of a subset of program participants.

These two descriptions make it unclear whether all interviewees were recruited from the existing study sample (which I imagine is the case). I recommend clarifying and ensuring consistency across both sections.

Qualitative Analysis (Line 255, p. 7)

Your description of the qualitative analysis would benefit from greater specificity. It is not clear whether your analytic logic was inductive, deductive, abductive, or retroductive. Additionally, it would be useful to specify the analytic approach employed, for example, whether this was content analysis, thematic analysis, or another qualitative approach to analysis. Providing this information will enhance the transparency and methodological robustness of the QUAL section of this study.

Results

The results section is clearly presented and easy to follow.

Additional comment: Your findings resonate with broader critiques regarding the scaling-up of interventions within the Global Mental Health field. The low attendance observed in the intervention itself is already telling. Even when strategies to increase attendance are implemented, they often aim to sustain research programs rather than reflect priorities co-constructed with participants and communities.I appreciate your discussion of strategies to improve adherence, including the use of financial incentives, especially given that participants’ productive time is already limited by conditions of precarity. At the same time, incentivization raises important questions about the potential for coercion, especially when structural needs remain unaddressed and the primary goal becomes maintaining engagement with programs that were not necessarily shaped by community priorities. To be clear, this is not a critique of your specific study but rather a reflection on broader trends within Global Mental Health intervention scaling up. If this view resonates with you, I would recommend adding a brief discussion on this in the discussion section.

---

## [Editor Report]

Thank you for submitting your manuscript to Global Mental Health. Overall, I agree with the reviewers that the manuscript is well-written, methodologically rigorous, and relevant to humanitarian and global mental health audiences. The reviewers particularly valued the mixed-methods design, the focus on displaced adolescent girls, and the integration of implementation science concepts. There were several consistent and important points made by reviewers that I encourage you to consider for revision: 1) providing more justification of why SSAGE was selected for this population and context, as well as some additional details about the intervention and implementers; 2) clarify some details about recruitment procedures; 3) explain how missing data were handled; and 4) report the internal consistency of measurement tools (e.g., RCADS) within this sample and consider adding some information about their relevance to the study setting/population. This is an excellent paper and I hope you will consider resubmitting a revised version to Global Mental Health.

---

## [Reviewer Report]

Thank you to the authors for receiving and addressing my concerns. I have no further comments. I congratulate the authors on their work.